# Diabetes mellitus and its association with central obesity, and overweight/obesity among adults in Ethiopia. A systematic review and meta-analysis

**Temesgen Muche Ewunie**[1]\* , **Daniel Sisay**[2], **Robel Hussen Kabthymer**[1]

1 Department of Human Nutrition, College of Health Sciences and Medicine, Dilla University, Dilla, Ethiopia,
2 Department of Public Health, College of Health Sciences and Medicine, Dilla University, Dilla, Ethiopia

\* temesgenmuche5270@gmail.com

## Abstract

### Background

Nowadays, diabetes mellitus is a serious public health problem in Ethiopia that has a profound impact on the health care system. However, no systematic synthesis and meta-analysis has been performed to depict the national prevalence. Hence, we authors aimed to assess the pooled prevalence of diabetes mellitus and its association with central obesity, overweight/obesity among adults in Ethiopia.

### Methods

We did a systematic review and meta-analysis of 15 eligible studies on the national prevalence of DM and its association with central obesity, and overweight/obesity among adults in Ethiopia. We searched PubMed/Medline, Science Direct, Embase, and Google Scholar, from August 01 up to October 28, 2021, in accordance with PRISMA guidelines. Joanna Briggs Institute (JBI) critical appraisal tool was used to assess the quality of studies. Analysis was done using STATA version 14 software. Heterogeneity was checked using the I-squared test, and the publication bias was examined by funnel plot and eggers test. Moreover, Sensitivity analysis was done to check the influence of small studies on the outcome. The trim and fill analysis was performed to estimate the potentially missing articles because of publication bias.

### Result

Total of 15 studies that met the inclusion criteria were included and the pooled prevalence of diabetes mellitus of the Federal Democratic Republic of Ethiopia was 6.26 (95%CI: 4.74–7.78). In the subgroup analysis, the prevalence of diabetes mellitus among the studies conducted in 2017 and before was 4.56 (95%CI: 2.98–6.14) but in studies done after 2017 was 7.55(95%CI: 4.69–10.41). The burden of diabetes mellitus was 5.79 times higher among those adults who had central obesity (OR = 5.79; 95%CI; 3.14–10.70), 5.70 times higher among adults who had overweight/obesity (OR = 5.70, 95%CI: 3.35–9.70).

**Data Availability Statement:** All relevant data are within the manuscript and its Supporting information files.

**Funding:** The authors received no specific funding for this work.

**Competing interests:** The authors have declared that no competing interests exist.

## Conclusion

The national prevalence of diabetes mellitus among adults in Ethiopia was higher and associated with central obesity, and overweight/obesity. Hence, the government of Ethiopia and stakeholders should give attention to strengthen the current health system regarding non-communicable diseases like diabetes mellitus and obesity/overweight.

## Introduction

Diabetes mellitus (DM) is a metabolic disease characterized by prolonged hyperglycemia due to either inadequate production of insulin by the pancreas or the cells of the body not responding properly to the produced insulin [1]. It is a major public health problem worldwide [2] and is largely associated with lifestyle changes [3].

Globally, the estimated prevalence of diabetes increases from time to time. It reached 463 million people in 2019, and this will be five hundred seventy-eight million, seven hundred million in 2030, and 2045, respectively, due to the current projection. Furthermore, the problem is worse in urban and high-income countries than in rural and low-income countries, with different consequences for the health, socioeconomic, and productivity of countries in general and people in particular [4].

The increment of diabetes mellitus prevalence is now becoming more significant in developing countries than in developed countries, where there are scarce resources for diabetic management, contributing to an increased risk of premature morbidity and mortality with major social and economic consequences [5]. The prevalence of diabetes has been steadily increasing over the past few decades. For instance, raised blood glucose is a common effect of uncontrolled diabetes and may, over time, lead to serious damage to the heart, blood vessels, eyes, kidneys, and nerves [4, 5].

Diabetes is a significant burden on the health care system and the economy at the national level in Sub–Saharan African countries, with the five leading countries with diabetes in 2017 being Ethiopia, South Africa, the Democratic Republic of the Congo, Nigeria, and Tanzania. Furthermore, the Sub–Saharan region is at high risk of receiving the highest percentage of cases of diabetes in any region in the world, so diabetes is a significant burden on the health care system and the economy at the national level [6].

The Ethiopian Diabetes Association (EDA) estimated a 2–3% prevalence in 2013 in Ethiopia. In 2015, the EDA (1.33 million) and IDF (1.30 million) reported almost the same number of people living with diabetes in the country [7].

Nowadays, the burden of diabetes mellitus has been increasing radically. The impact is high, especially in Ethiopia, where resources are limited to identifying the problem and developing need-based clinical and community intervention.

This burden can be measured through direct medical costs, indirect costs associated with productivity loss, premature mortality, and the negative impact of diabetes on nations' gross domestic product (GDP). Therefore, a systematic review and meta-analysis is needed that shows the burden of DM at the national level and its association with central obesity, and overweight and obesity among adult in Ethiopia.

## Materials and methods

### Search strategy and review process

The authors (TME and DS) conducted a comprehensive search using electronic databases (PubMed/Medline, Science Direct, and Embase) and manual search (Google Scholar) from August 1 up to October 28, 2021. This review was conducted according to Preferred Reporting Items for Systematic Reviews and Meta-Analyses (PRISMA 2020) guidelines [8]. We authors, used the following keywords during our search; ((Prevalence) OR (burden)) OR (epidemiology)) OR (level)) AND (associated factors)) OR (determinants)) AND (diabetes mellitus)) AND (adult)) AND (Ethiopia). Then this systematic review and meta-analysis was performed in accordance with The Preferred Reporting Items for Systematic Reviews and Meta-Analyses checklist for reporting a systematic review or meta-analysis protocol (Fig 1).

### Eligibility criteria

1. **Study area**: Research articles conducted only in Ethiopia were included.

2. **Study design:** Observational studies (cross-sectional, case-control, and cohort studies) with original data reporting the prevalence of DM and its associated factors were included.

3. **Language**: Literatures published in the English language were considered.

4. **Population:** Study conducted on adults population were included.

5. **Publication issue:** Both published and unpublished articles were included in this review.

6. **Study period:** Study with no time limit on study period were included.

7. Studies that do not report the prevalence of diabetes mellitus were excluded.

**Data extraction.**   A reviewer (T.M.E) extracted the data using a standard Microsoft excel sheet adapted using the Joanna Briggs Institute (JBI) quality score [9]. And the name of the first author, publication year, region, setting, quality score, sample size, prevalence, and diagnostic criteria were considered in the extraction process. The second reviewer (D.S) revised the extracted data.

**Data processing and analysis.**   In this systematic review and meta-analysis, the data was extracted using standard Microsoft excel format and then exported into STATA version 14 software for analysis. We authors used the random-effects model were used to pool outcome results from eligible studies. The pooled prevalence of the outcome variable with 95% confidence interval was reported. Heterogeneity was checked using the I-squared test. To assess the publication bias we used funnel plot observation subjectively, Begg test, and egger's test. Statistically, publication bias was declared at a p-value less than 0.05. Sensitivity analysis (leave-one-out) was done to test the influence of a single study on the pooled prevalence, by assuming zero differences between groups. The trim and fill analysis was performed to estimate the potentially missing articles because of publication bias. And sub-group analysis was performed by study period, region, and sample size.

## Results

Fifteen studies [7, 10–22], with a total of 13,774 adults, met the inclusion criteria (Fig 1), and a total of 13,774 adults aged 15 years and above were included in this review. Also, articles were identified using an electronic database and a manual search for the prevalence of diabetes

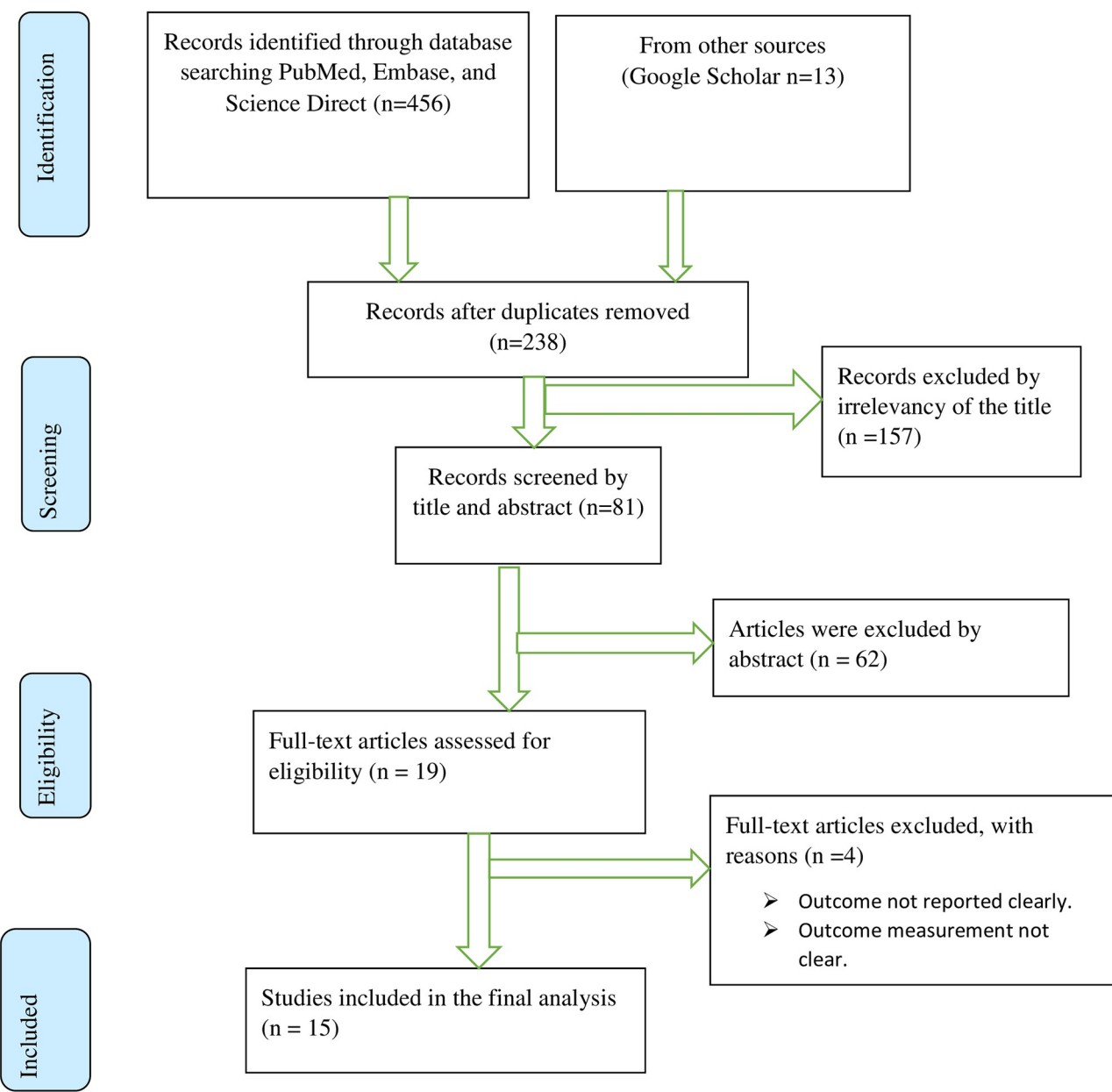

**Fig 1. A PRISMA flow chart for systematic review and meta-analysis.**

mellitus among adults in Ethiopia. All the studies were observational (cross-sectional studies) and the smallest sample size was 392 in Gondar Ethiopia [18], while the largest sample size was 2922 in Bona district, SNNPR [7] participants. Studies were conducted in different parts of Ethiopia, 3 were conducted in the southern region [7, 15, 22], 6 conducted in Amhara [10, 12, 13, 17, 18, 20], 4 conducted in Addis Ababa [11, 13, 19, 21] and 2 in the Sidama region [14, 23]. The sample size of the included studies was ranging from 1.9% [7] to 14.8% [11] (Table 1).

**Table 1. Summary of included studies to assess the pooled prevalence of diabetes mellitus and its association with central obesity and overweight/obesity in Ethiopia.**

| Author | Publication Year | Region | Setting | Quality Ass.(JBI) | Sample Size | Outcome | Prevalence | Diagnostic Criteria |
|---|---|---|---|---|---|---|---|---|
| Sahile and Bekele [11] | 2020 | AA | Urban | 7 | 758 | 112 | 14.8 | Not stated |
| Endris et al. [12] | 2019 | Amhara | Urban | 6 | 587 | 40 | 6.8 | ADA |
| Seifu et al. [23] | 2020 | Sidama | Both | 7 | 519 | 64 | 12.4 | WHO |
| Dereje N, et al. [22] | 2020 | SNNPR | Urban | 6 | 634 | 36 | 5.7 | WHO |
| Aynalem and Zeleke [15] | 2018 | SNNPR | Urban | 5 | 414 | 26 | 6.5 | ADA |
| Alemayehu Z. et al. [7] | 2018 | SNNPR | Both | 6 | 2922 | 51 | 1.9 | ADA |
| Woldesemayat et al. [13] | 2019 | AA | Urban | 7 | 422 | 10 | 2.6 | WHO |
| Kassa A. and Woldesemayat E. [14] | 2019 | Sidama | Urban | 7 | 423 | 50 | 12.2 | Not stated |
| Abebe et al. [20] | 2014 | Amhara | Both | 8 | 2200 | 77 | 3.6 | WHO |
| Tesfaye et al. [19] | 2016 | AA | Urban | 7 | 1003 | 47 | 5 | Not stated |
| Wolde et al. [10] | 2020 | Amhara | Both | 7 | 805 | 49 | 6.3 | WHO |
| Uhomoibhi p. [21] | 2003 | AA | Urban | 5 | 533 | 16 | 3.4 | WHO |
| Animaw W and Seyoum y. [16] | 2017 | Amhara | Both | 6 | 1405 | 46 | 3.3 | Not stated |
| Wondemagegn et al. [17] | 2017 | Amhara | Both | 7 | 757 | 83 | 11 | WHO |
| Worede et al. [18] | 2017 | Amhara | Urban | 7 | 392 | 9 | 2.3 | ADA |

Notice: AA; Addis Ababa, ADA; American Diabetic Association, WHO; World Health Organization

## Meta-analysis of diabetes mellitus among adults

The pooled prevalence of diabetes mellitus in Ethiopia was found to be 6.26% (95%CI: 4.74–7.78). The heterogeneity of the pooled estimate ($I^2$ = 94.6%, P = 0.000) and a random-effects model was used to decrease heterogeneity (Fig 2).

## Publication bias

Publication bias was checked using a funnel plot and objectively by the eggers test. We found in this study publication bias, as evidenced by substantial asymmetric funnel plot (Fig 3) and statistically significant begg test (P = 0.001) (S1 Fig) and egger's test (P = 0.000) (S2 Fig). In addition to this, the sensitivity analysis finding revealed that the studies had no effect on the pooled prevalence of diabetes mellitus among adults in Ethiopia (Fig 4).

## Subgroup analysis

In the subgroup analysis, the pooled prevalence of diabetes mellitus was 4.56% from the studies conducted in 2017 and before. Whereas, the prevalence of DM was 7.55 in the studies carried out after 2017 (Table 2).

## Association of diabetes mellitus with overweight/obesity

In this review, participants with overweight/obesity reported from nine studies [7, 11–15, 19, 22, 23] and the odds of diabetes mellitus was 5.70 times higher among adults who had overweight/obesity than their counterparts (OR = 5.70, 95%CI: 3.35–9.70) (Fig 5).

## Association of diabetes mellitus with central obesity

The association between diabetes mellitus and central obesity was computed from six studies [7, 12, 13, 15, 19, 23]. The prevalence of diabetes mellitus was 5.79 times higher among those

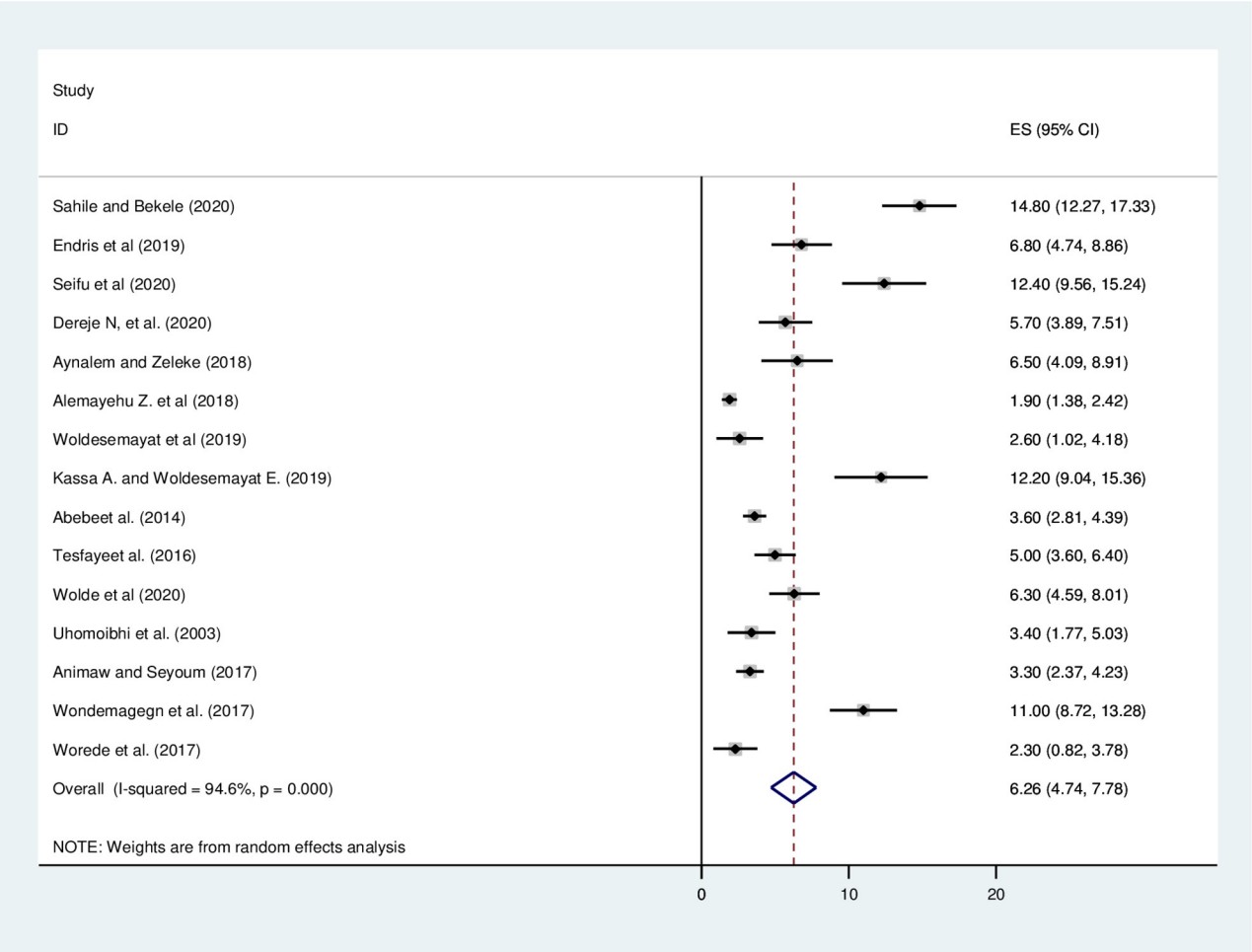

**Fig 2. Forest plot of fifteen studies which included to assess the pooled prevalence of diabetes mellitus in Ethiopia.**

adults who had central obesity than those who did not have central obesity (OR = 5.79; 95%CI; 3.14–10.70) (Fig 6).

## Discussion

In this systematic review and meta-analysis and a total of 13,774 study participants were included from fifteen eligible studies [7, 10–23]. The pooled prevalence of diabetes mellitus among adults in Ethiopia was found to be 6.26%. This is higher than the study conducted in Nigeria, 3% [24]. But lower than the studies conducted in developed countries, like Germany, 14% [25], Thailand 16.8% [26], and Belgium 9.4% [27]. This could be due to different reasons such as sedentary lifestyles and urbanization in developed countries. And consistent with the study conducted in New Zealand 6% [28].

The current review showed that there is an increment in the prevalence of DM from time to time. The present finding is supported by different the studies [29, 30]. This might be an increment of unhealthy dietary behaviour like high fat diet consumption, physical inactivity, and urbanization, which are the risk factors for non-communicable diseases including diabetes mellitus.

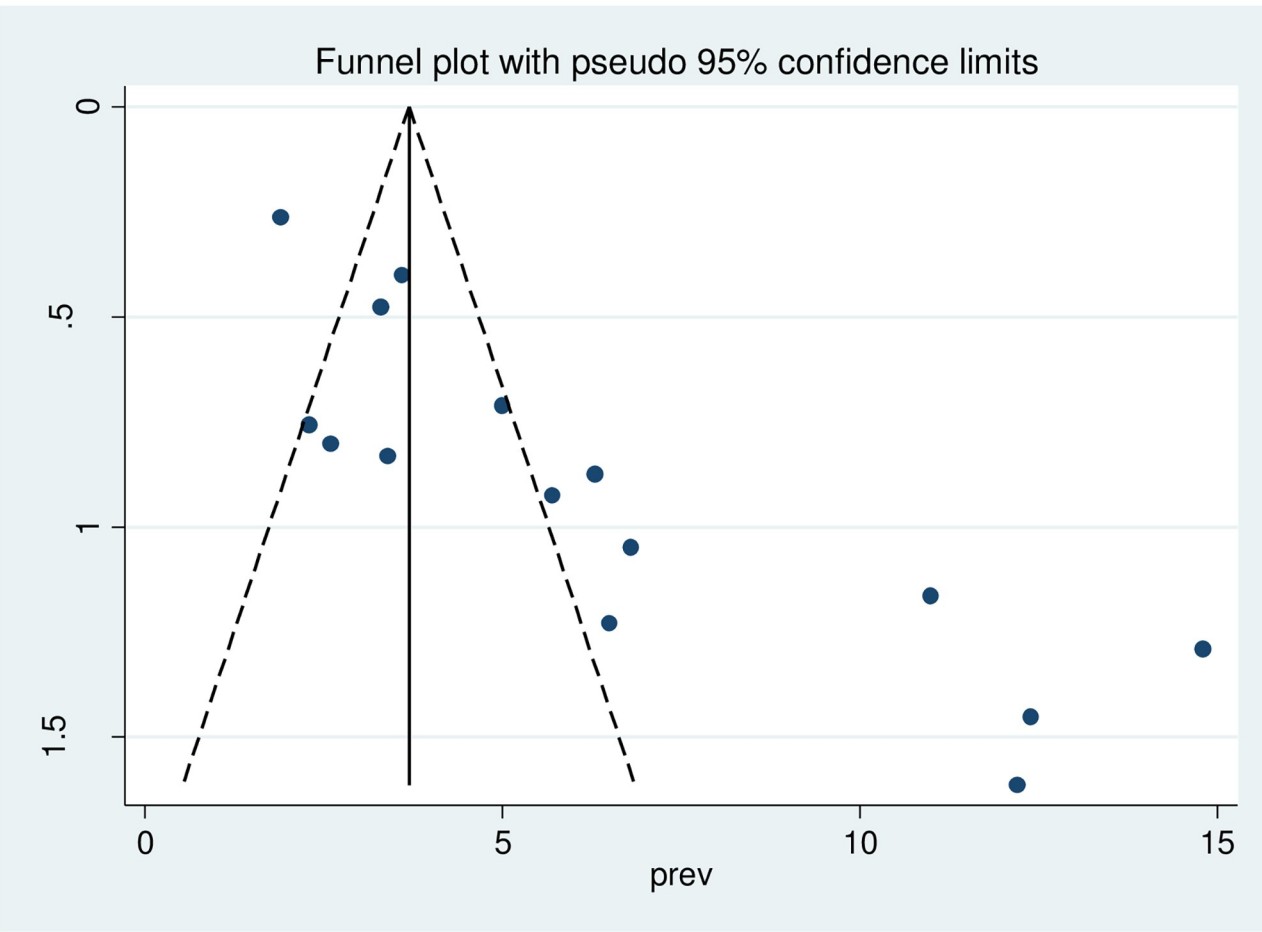

**Fig 3. Funnel plot test output of the included studies.**

In addition, we observed that diabetes mellitus was positively associated with increased odds of having obesity/overweight and central obesity. This finding is consistent with the study conducted in China [31]. In this review, a high degree of heterogeneity was observed. This might be due to differences in the categorization of overweight/obesity. Although the pathophysiology of the relationship between diabetes mellitus and central obesity, overweight/obesity is still not well known, having central obesity and/or being overweight make the treatment/control of diabetes mellitus difficult among DM patients.

This meta-analysis has several strengths. To our knowledge, it is the first review that combined fifteen primary studies and provide up-to-date data that showed the national burden of diabetes in Ethiopia. In spite of these strengths, the review has some limitations, such as included studies were cross-sectional, which could not show the causal association between diabetes mellitus with abdominal obesity and overweight/obesity. Despite the estimated burden of diabetes mellitus and its association with obesity, we are unable to assess the pathophysiology between DM and obesity. Because of inadequate primary studies, the review was conducted based on the studies conducted in four regions of the country from a total of ten regions which limits the generalizability of the findings at the national level. Furthermore, although different efforts have been made during analysis to reduce the effect of bias, the

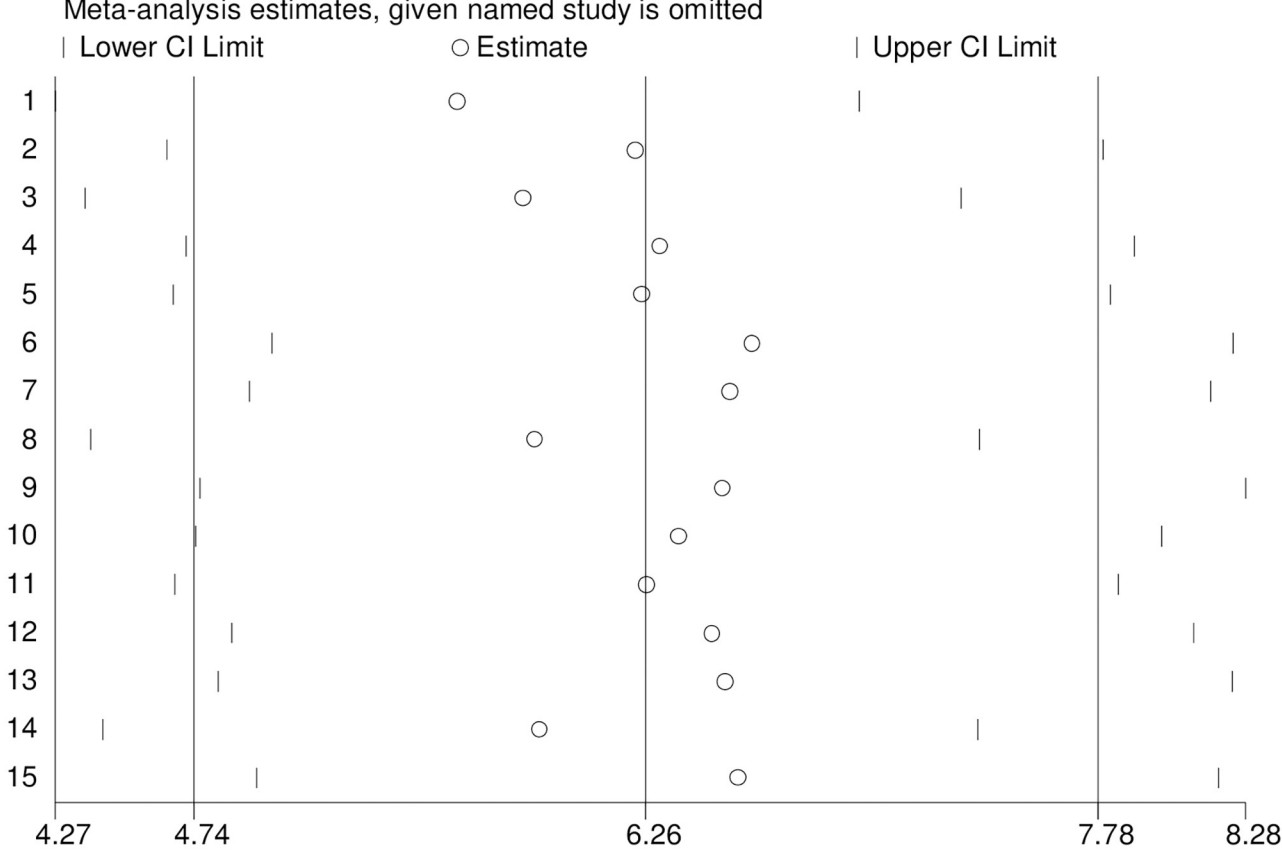

**Fig 4. Output of sensitivity analysis of 15 studies.**

presence of publication bias is also the other limitation of this meta-analysis. Therefore, a follow-up study should be conducted in order to confirm the association between the outcome variable and abdominal obesity, overweight/obesity in the future.

## Conclusion

The current review found evidence that the prevalence of diabetes mellitus among adults was dramatically increasing from time to time. It has a strong association with central obesity, and overweight/obesity. To mitigate this health challenge, it is necessary to integrate control

**Table 2. Subgroup analysis of diabetes mellitus by publication year and sample size of studies conducted in Ethiopia.**

| | Category | Number of studies | ES(95%CI) | I-Squared (%) |
|---|---|---|---|---|
| Region | Addis Ababa | 4 | 6.33(2.21–10.46) | 95.7 |
| | Amhara | 6 | 5.35(3.46–7.22) | 91.4 |
| | Sidama | 2 | 12.31(10.20–14.43) | 0.0 |
| | SNNPR | 3 | 4.56(1.26–7.84) | 92.7 |
| Publication year | 2017 and before | 6 | 4.56(2.98–6.14) | 89.2 |
| | After 2017 | 9 | 7.55(4.69–10.41) | 96.2 |
| Sample size | ≥700 | 7 | 6.29(4.11–8.46) | 96.5 |
| | <700 | 8 | 6.27(3.97–8.57) | 90.8 |

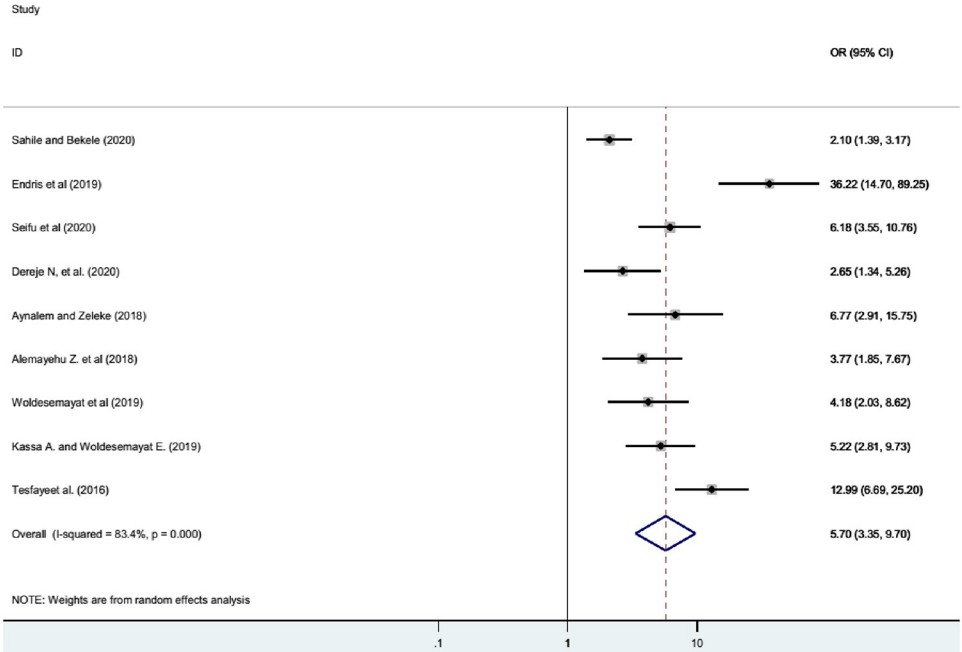

**Fig 5. Forest plot showing the association between DM and overweight/obesity in Ethiopia.**

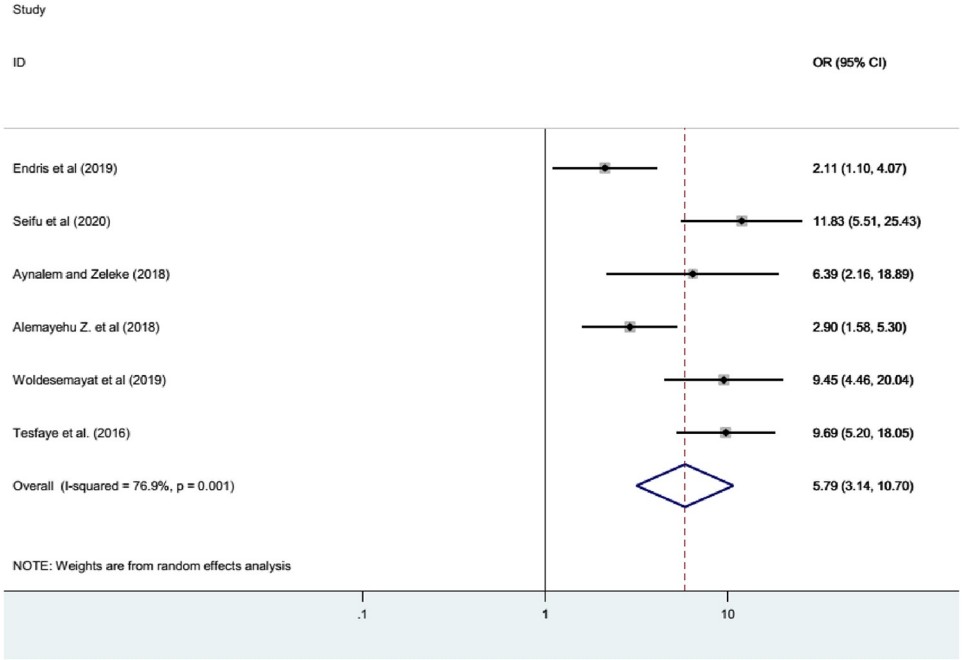

**Fig 6. Forest plot showing the association between DM and central obesity in Ethiopia.**

strategies with other health services, promote nutritional intervention, and encourage physical activity.

## Supporting information

**S1 Table. PRISMA 2020 checklist.**
(DOCX)

**S2 Table. Literature screening Microsoft excel sheet.**
(DOCX)

**S3 Table. JBI-quality assessment tool for cross-sectional studies.**
(DOCX)

**S1 Fig. Begg's test analysis for diabetes mellitus and its association with central obesity, and overweight/obesity among adults.**
(TIFF)

**S2 Fig. Egger's test result for diabetes mellitus and its association with central obesity, and overweight/obesity among adults.**
(TIFF)

## Acknowledgments

We would like to thank Dilla University for supporting this systematic review and meta-analysis.

## Author Contributions

**Conceptualization:** Temesgen Muche Ewunie.

**Data curation:** Temesgen Muche Ewunie, Daniel Sisay, Robel Hussen Kabthymer.

**Formal analysis:** Temesgen Muche Ewunie, Robel Hussen Kabthymer.

**Investigation:** Temesgen Muche Ewunie.

**Methodology:** Temesgen Muche Ewunie, Daniel Sisay.

**Software:** Temesgen Muche Ewunie, Robel Hussen Kabthymer.

**Validation:** Temesgen Muche Ewunie.

**Visualization:** Daniel Sisay.

**Writing – original draft:** Temesgen Muche Ewunie.

**Writing – review & editing:** Temesgen Muche Ewunie, Daniel Sisay, Robel Hussen Kabthymer.

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
