## [Decision Letter · Decision Letter 0]

13 May 2022

PONE-D-22-05097Diabetes Mellitus and its Association with Central Obesity, Overweight/Obesity among adults in Ethiopia. A Systematic Review and Meta-AnalysisPLOS ONE

Dear Dr. Muche,

Thank you for submitting your manuscript to PLOS ONE. After careful consideration, we feel that it has merit but does not fully meet PLOS ONE’s publication criteria as it currently stands. Therefore, we invite you to submit a revised version of the manuscript that addresses the points raised during the review process.

Plesse address all reviewer's comments.

We look forward to receiving your revised manuscript.

Kind regards,

Paolo Magni

Academic Editor

PLOS ONE

Journal Requirements:

2. Please upload a new copy of Figure 5 as the detail is not clear. Please follow the link for more information: " ext-link-type="uri" xlink:type="simple">https://blogs.plos.org/plos/2019/06/looking-good-tips-for-creating-your-plos-figures-graphics/"
https://blogs.plos.org/plos/2019/06/looking-good-tips-for-creating-your-plos-figures-graphics/

“NO: The funders had no role in study design, data collection and analysis, decision to publish, or preparation of the manuscript”

Additional Editor Comments (if provided):

please see reviewer's comments

Reviewers' comments:

Reviewer's Responses to Questions

**Comments to the Author**

1. Is the manuscript technically sound, and do the data support the conclusions?

Reviewer #1: Partly

2. Has the statistical analysis been performed appropriately and rigorously? 

Reviewer #1: No

3. Have the authors made all data underlying the findings in their manuscript fully available?

Reviewer #1: Yes

4. Is the manuscript presented in an intelligible fashion and written in standard English?

Reviewer #1: No

5. Review Comments to the Author

Reviewer #1: Dear Editor,

I carefully read the manuscript by Muche et al.

My comments and suggestions are the following:

- Line 125: The authors referred to an obsolete version of PRISMA guidelines (Ref. 8). They should revise the analysis following the updated PRISMA guidelines (that have been published early last year).

- Line 152: The authors should also perform Begg test.

- Line 152-153: The authors should specify how sensitivity analysis was performed.

- Table 1: Information regarding the prevalence by sex should be included in the table.

- The authors should consider to refer to doi: 10.1016/j.numecd.2020.03.005 in their manuscript.

- The limitations of the analysis should be more deeply discussed.

- English language needs to be revised.

6. PLOS authors have the option to publish the peer review history of their article (what does this mean?). If published, this will include your full peer review and any attached files.

Reviewer #1: No

---

## [Author Response · Author response to Decision Letter 0]

24 May 2022

Reviewer #1 

Reviewers comment 

1 Line 125: The authors referred to an obsolete version of PRISMA guidelines (Ref. 8). They should revise the analysis following the updated PRISMA guidelines (that have been published early last year). Thank you very much for your suggestion. It has been corrected based on the recent PRISMA-2020 guidelines” page 13

2 Line 152: The authors should also perform Begg test. Thank you very much for your insightful comments. As per the reviewer’s comment it has been corrected. Page 9 

3 Line 152-153: The authors should specify how sensitivity analysis was performed. Thanks! The authors addressed the given comments. Page 7

4 Table 1: Information regarding the prevalence by sex should be included in the table. Thanks for your insightful comments. It would have been better if we could have this variable, but most of the eligible studies lack this variable. So, we are unable to assess the prevalence based on sex.

5 The authors should consider to refer to doi: 10.1016/j.numecd.2020.03.005 in their manuscript. Thanks for the comments. We found this literature crucial for our review. Page 11 and reference number 30.

6 The limitations of the analysis should be more deeply discussed. As per the comment it has be discussed Page 12

7 English language needs to be revised. Thanks!. We authors made a correction as per the given comment. Throughout the document

---

## [Decision Letter · Decision Letter 1]

30 May 2022

Diabetes mellitus and its association with central obesity, and overweight/obesity among adults in Ethiopia. A systematic review and meta-analysis

PONE-D-22-05097R1

Dear Dr. Temesgen Muche,

We’re pleased to inform you that your manuscript has been judged scientifically suitable for publication and will be formally accepted for publication once it meets all outstanding technical requirements.

Kind regards,

Paolo Magni

Academic Editor

PLOS ONE

Additional Editor Comments (optional):

The paper has been significantly improved.

Reviewers' comments:

Reviewer's Responses to Questions

**Comments to the Author**

1. If the authors have adequately addressed your comments raised in a previous round of review and you feel that this manuscript is now acceptable for publication, you may indicate that here to bypass the “Comments to the Author” section, enter your conflict of interest statement in the “Confidential to Editor” section, and submit your "Accept" recommendation.

Reviewer #1: All comments have been addressed

2. Is the manuscript technically sound, and do the data support the conclusions?

Reviewer #1: Yes

3. Has the statistical analysis been performed appropriately and rigorously? 

Reviewer #1: Yes

4. Have the authors made all data underlying the findings in their manuscript fully available?

Reviewer #1: Yes

5. Is the manuscript presented in an intelligible fashion and written in standard English?

Reviewer #1: Yes

6. Review Comments to the Author

Reviewer #1: I carefully read the revised version of the manuscript that is significantly improved in comparison with the previous version.

7. PLOS authors have the option to publish the peer review history of their article (what does this mean?). If published, this will include your full peer review and any attached files.

Reviewer #1: No

---

## [Editor Report · Acceptance letter]

2 Jun 2022

PONE-D-22-05097R1 

Diabetes mellitus and its association with central obesity, and overweight/obesity among adults in Ethiopia. A systematic review and meta-analysis 

Dear Dr. Muche:

I'm pleased to inform you that your manuscript has been deemed suitable for publication in PLOS ONE. Congratulations! Your manuscript is now with our production department. 

Kind regards, 

on behalf of

Prof. Paolo Magni 

Academic Editor

PLOS ONE